# α_2_δ-4 and Cachd1 Proteins Are Regulators of Presynaptic Functions

**DOI:** 10.3390/ijms23179885

**Published:** 2022-08-31

**Authors:** Cornelia Ablinger, Clarissa Eibl, Stefanie M. Geisler, Marta Campiglio, Gary J. Stephens, Markus Missler, Gerald J. Obermair

**Affiliations:** 1Institute of Physiology, Medical University Innsbruck, 6020 Innsbruck, Austria; 2Division Physiology, Department of Pharmacology, Physiology and Microbiology, Karl Landsteiner University of Health Sciences, 3500 Krems, Austria; 3Department Pharmacology and Toxicology, University of Innsbruck, 6020 Innsbruck, Austria; 4Reading School of Pharmacy, University of Reading, Reading RG6 6UB, UK; 5Institute of Anatomy and Molecular Neurobiology, Westfälische Wilhelms-University, 48149 Münster, Germany

**Keywords:** α_2_δ subunits, Cachd1, synapse formation, synaptic differentiation, presynaptic calcium imaging, voltage-gated calcium channels

## Abstract

The α_2_δ auxiliary subunits of voltage-gated calcium channels (VGCC) were traditionally regarded as modulators of biophysical channel properties. In recent years, channel-independent functions of these subunits, such as involvement in synapse formation, have been identified. In the central nervous system, α_2_δ isoforms 1, 2, and 3 are strongly expressed, regulating glutamatergic synapse formation by a presynaptic mechanism. Although the α_2_δ-4 isoform is predominantly found in the retina with very little expression in the brain, it was recently linked to brain functions. In contrast, Cachd1, a novel α_2_δ-like protein, shows strong expression in brain, but its function in neurons is not yet known. Therefore, we aimed to investigate the presynaptic functions of α_2_δ-4 and Cachd1 by expressing individual proteins in cultured hippocampal neurons. Both α_2_δ-4 and Cachd1 are expressed in the presynaptic membrane and could rescue a severe synaptic defect present in triple knockout/knockdown neurons that lacked the α_2_δ-1-3 isoforms (α_2_δ TKO/KD). This observation suggests that presynaptic localization and the regulation of synapse formation in glutamatergic neurons is a general feature of α_2_δ proteins. In contrast to this redundant presynaptic function, α_2_δ-4 and Cachd1 differentially regulate the abundance of presynaptic calcium channels and the amplitude of presynaptic calcium transients. These functional differences may be caused by subtle isoform-specific differences in α_1_-α_2_δ protein–protein interactions, as revealed by structural homology modelling. Taken together, our study identifies both α_2_δ-4 and Cachd1 as presynaptic regulators of synapse formation, differentiation, and calcium channel functions that can at least partially compensate for the loss of α_2_δ-1-3. Moreover, we show that regulating glutamatergic synapse formation and differentiation is a critical and surprisingly redundant function of α_2_δ and Cachd1.

## 1. Introduction

Voltage-gated calcium channels (VGCC) are multi-subunit protein complexes consisting of a pore-forming α_1_ and the auxiliary β and α_2_δ subunits, which are traditionally regarded as modulators of biophysical channel properties [1,2,3]. In the human genome, four genes (CACNA2D1-4) encode for four α_2_δ proteins (α_2_δ-1 to α_2_δ-4), which all are composed of a large glycosylated α_2_ peptide that is likely linked to a GPI membrane-anchored δ peptide [1,4]. In recent years, a number of studies have linked human α_2_δ genes to neurological and neuropsychiatric disorders, such as epilepsy, autism, schizophrenia, and anxiety disorders (reviewed [5,6]). Moreover, α_2_δ proteins were recently recognized as important regulators of synaptic functions, some of which may be independent of calcium channels. For example, α_2_δ-1 has been identified as a postsynaptic receptor for the glia-secreted thrombospondins which mediate excitatory synapse formation [7,8]. Loss of α_2_δ-2 was shown to affect cerebellar climbing fiber synapses [9] and postsynaptic calcium signaling function of cerebellar Purkinje neurons [10,11,12], as well as the structure and function of auditory hair cell synapses [10]. Moreover, in CNS neurons, α_2_δ-2 functions as a trans-synaptic organizer, recruiting postsynaptic GABA_A_ receptors [11]. α_2_δ-3 is important for the formation of auditory nerve fiber synapses in mice [12], while invertebrate α_2_δ-3 homologues are required for the presynaptic development of motoneurons [13,14]. In contrast to α_2_δ-1, -2 and -3, which are highly expressed in the brain, the fourth isoform, α_2_δ-4, only shows very low expression levels in the cortex and hippocampus [15]. However, α_2_δ-4 is specifically expressed in the photoreceptor cells of the retina, where it mainly interacts with the Ca_V_1.4 presynaptic L-type channel, and, importantly, is required for the proper organization of rod and cone photoreceptor synapses [16,17,18,19]. Recent studies also suggested a functional role of lowly expressed α_2_δ-4 in the brain. mRNA levels of α_2_δ-4 were increased in human hippocampal biopsies obtained from epileptic patients [20], and a recent study on knockout mice identified a role for α_2_δ-4 in cognitive and motor behavior [21].

A putative cache (Ca^2+^ channel and chemotaxis receptor) domain containing protein 1 (CACHD1) has been described as an α_2_δ-like protein, displaying structural similarities to members of the α_2_δ family and showing a wide expression in the CNS [22,23]. An expression study using rat and zebrafish illustrated that Cachd1 increases N-type calcium currents and surface expression and competes with α_2_δ-1 for binding to Ca_V_2.2 [24]. Interestingly, upon co-expression, human CACHD1 also increased cell-surface localization of the Ca_V_3.1 T-type channel [22], although T-type channels are not known to interact with auxiliary β and α_2_δ subunits [1]. Furthermore, overexpression of CACHD1 in hippocampal neurons caused a pronounced increase in T-type current-mediated spike firing [23]. The question of whether and how Cachd1 and α_2_δ-4 contribute to specific functions in neurons, which simultaneously express all α_2_δ isoforms, or whether they share functional redundancy with other α_2_δ isoforms, remains to be answered. Hence, considering that all α_2_δ subunits, including α_2_δ-4, have been linked to synapse formation, it is tempting to speculate that organizing synapse formation is a general feature of α_2_δ proteins.

By establishing an α_2_δ triple knockout/knockdown (TKO/KD) neuronal model system, we showed that the loss of presynaptic α_2_δ proteins leads to a severe defect in glutamatergic synapse formation and synaptic transmission. This striking phenotype could be rescued by the individual expression of the endogenous CNS isoforms α_2_δ-1, -2, and -3 [25]. In the present study, we aimed to examine the synaptic functions of α_2_δ-4 and the α_2_δ-like protein Cachd1. An analysis of surface expression revealed that α_2_δ-4 and Cachd1 can be targeted to the synaptic membrane, similar to α_2_δ-1 and α_2_δ-2, suggesting their involvement in specific synaptic functions. To investigate if mediating synaptogenesis is a general feature of α_2_δ proteins, we next examined the synaptogenic potential of α_2_δ-4 and Cachd1 in α_2_δ TKO/KD neurons. The expression of α_2_δ-4 and Cachd1 rescued synapse formation, presynaptic calcium channel clustering and presynaptic calcium transients in TKO/KD neurons. However, α_2_δ-4 and Cachd1 displayed subtle differences in their rescue abilities, suggesting distinct protein-specific synaptic functions. Indeed, α_2_δ-4 and Cachd1 differentially affected presynaptic calcium transients in wildtype neurons, where α_2_δ-4 showed reduced calcium signaling in comparison to Cachd1. Taken together, we provide the first evidence that α_2_δ-4 and Cachd1 have important synaptic roles in mediating glutamatergic synapse formation and differentially affecting presynaptic calcium transients.

## 2. Results

### 2.1. Epitope-Tagged α_2_δ-4 and Cachd1 Target to Presynaptic Terminals

The α_2_δ-like protein Cachd1 is strongly expressed in multiple brain areas including the hippocampus, cerebellum, and thalamus [22]. In contrast, CNS expression of α_2_δ-4 is very low in the hippocampus [15,20]. While the subcellular localization and synaptic roles of Cachd1 are largely unknown, the synaptic actions of α_2_δ-4 have been studied in photoreceptor cells using knockout mice [18]. Because neuronal α_2_δ proteins are critical regulators of synaptic transmission and synapse formation, in the present research, we studied whether Cachd1 and the retinal α_2_δ-4 subunit also are potential regulators of synaptic functions. As synaptic functions of proteins are normally associated with a synaptic localization, we first analyzed the subcellular distribution of Cachd1 and α_2_δ-4 by expressing HA-epitope-tagged proteins in primary cultured hippocampal neurons, as previously established [11,25,26,27]. To visualize the neuronal morphology, neurons were co-transfected with soluble eGFP [11,25,26,27]. Life cell labeling revealed that, similar to α_2_δ-1 and α_2_δ-2 isoforms [11,25], Cachd1 and α_2_δ-4 were strongly expressed on the surface of the cell soma (Figure 1A) and the dendrites including dendritic spines, opposite functional synapses which were identified by immunolabeling the presynaptic synapsin protein (Figure 1B, arrowheads). Quantification showed somato-dendritic expression levels of Cachd1 and α_2_δ-4 between the levels of α_2_δ-1 and α_2_δ-2, which showed the weakest and strongest surface expression, respectively, of all α_2_δ proteins (Figure 1C, soma and dendrite).

In general, the labelling intensities on the axonal surface were in line with the overall expression levels for all proteins (Figure 1C), although the axonal expression of Cachd1 was lower compared to those of soma and dendrites (Figure 1C, compare Cachd1 expression levels in soma, dendrite, and axon). Importantly, Cachd1 and α_2_δ-4 were also expressed in the presynaptic membrane (Figure 2). This is particularly evident in the upper panel of Figure 2, displaying HA labeling in the outlined presynaptic membrane (white dotted lines drawn around the eGFP signal). Remarkably, and like α_2_δ-2 [25], α_2_δ-4 showed a distinct peri-synaptic localization in the presynaptic membrane around the centrally located synapsin cluster (Figure 2, upper panel and linescan). Together, these data suggest that HA-tagged Cachd1 and α_2_δ-4 target presynaptic terminals upon expression in cultured hippocampal neurons, in line with previous suggestions of the CNS functions of both proteins [20,21,22].

### 2.2. α_2_δ-4 and Cachd1 Rescue Glutamatergic Synapse Formation

We previously established an α_2_δ triple knockout/knockdown model (TKO/KD) by combining classical double knockout (α_2_δ-3)/mutant (α_2_δ-2) mice with shRNA knockdown (α_2_δ-1) [25]. Presynaptic α_2_δ TKO/KD leads to a severe defect in glutamatergic synapse formation and synaptic transmission [25]. Importantly, this phenotype can be rescued by the individual expression of neuronal α_2_δ isoforms (α_2_δ-1, -2, and -3) [25]. Due to the severe synaptic defect, including the strong reduction of presynaptic synapsin and Ca_V_2.1 expression, presynaptic calcium transients, and synaptic transmission (Figure 3 and Figure 4), TKO/KD neurons served as an ideal model for studying the synaptic functions and synaptogenic potential of homologously expressed α_2_δ proteins. Because Cachd1 and α_2_δ-4 can both target presynaptic terminals (Figure 2), we next tested whether their expression could also rescue glutamatergic synapse formation and function in α_2_δ TKO/KD neurons. Expression of α_2_δ-4 and Cachd1 in TKO/KD neurons significantly rescues presynaptic synapsin and Ca_V_2.1 clustering to levels comparable to α_2_δ-1 (Figure 3A, representative images and line scans, Figure 3B,C). This suggests that α_2_δ-4 or Cachd1 can both rescue the severe TKO/KD phenotype by restoring the presynaptic expression of synaptic vesicle-associated proteins and calcium channels. This is in line with previous observations for α_2_δ-1, α_2_δ-2, and α_2_δ-3 isoforms [25].

### 2.3. α_2_δ-4 and Cachd1 Restore Presynaptic Calcium Signaling in Triple Knockout Neurons

Restoring presynaptic differentiation in α_2_δ TKO/KD neurons by the expression of α_2_δ-4 and Cachd1 suggests a presynaptic rescue of the severe phenotype. Hence, we next analyzed whether this rescue effect was also reflected in other synaptic functions. To this end, we used calcium imaging with the genetically encoded and presynaptically targeted calcium sensor SynGCaMP6f [25,28] to test if synaptic calcium signaling could be completely or partially rescued by α_2_δ-4 and Cachd1. In line with previous results, a significant fraction of TKO/KD synapses failed to show calcium transients in response to stimulations with 1 action potential (AP), 3 APs, and 10 APs (Figure 4, black lines and symbols). This was particularly evident when plotting the cumulative frequency distribution of single synaptic responses (Figure 4B), which illustrated that 56, 37, and 34% of TKO/KD synapses showed no response to 1, 3, or 10 AP stimulations, respectively. In contrast, only 26% (1 AP), 12% (3 APs), and 5% (10 APs) of control synapses were non-responding (Figure 4, green lines and symbols). In line with previous observations, the expression of α_2_δ-1 rescued the peak amplitude of presynaptic calcium transients (Figure 4, orange lines). This was evident when analyzing the peak responses of all individual synapses (Figure 4B), as well as when plotting the means of synapses from all individual cells analyzed (Figure 4C). The expression of either α_2_δ-4 (Figure 4, blue lines) or Cachd1 (Figure 4, magenta lines) rescued presynaptic calcium transients to levels comparable to control (Figure 4, green lines) and α_2_δ-1 (Figure 4, orange lines), as evidenced by the similar means of all individual cells under all tested stimulation patterns (Figure 4C). Although not evident in the analysis of the means of all individual cells (Figure 4C), the peak amplitudes of presynaptic calcium transients of TKO/KD cells expressing α_2_δ-4 seemed to be slightly smaller compared to the rescue conditions with α_2_δ-1 and Cachd1 (Figure 4A, compare mean trances; Figure 4B, cumulative frequency distribution,). Furthermore, the fraction of non-responding synapses was significantly reduced (Chi-square test 1 AP: χ^2^ = 130.9, *p* < 0.0001; 3 AP: χ^2^ = 34.1, *p* < 0.0001; 10 AP: χ^2^ = 38.4, *p* < 0.0001) upon expression of Cachd1. For example, upon 1 AP stimulation, only 14% of synapses expressing Cachd1 were non-responding, while in synapses expressing α_2_δ-1 and α_2_δ-4, the non-responding fraction accounted for 37% and 45%, respectively (Figure 4B). A similar albeit not significant trend was seen upon stimulation with 3 and 10 APs. Taken together, these results demonstrate that all α_2_δ subunits and α_2_δ-like proteins could rescue presynaptic protein expression and calcium signaling in TKO/KD neurons, suggesting that they share redundant functions in synapse formation and differentiation. Moreover, the subtle differences between Cachd1 and α_2_δ-4 in terms of rescuing presynaptic calcium transients may be related to differences in their abilities to interact with presynaptic calcium channels.

### 2.4. α_2_δ-4 and Cachd1 Do Not Increase Presynaptic Ca_V_2.1 Abundance 

The above data suggest that expression of individual α_2_δ subunits and Cachd1 is sufficient for the proper differentiation and function of glutamatergic synapses. Nevertheless, the differential effects on presynaptic calcium transients may suggest a subtle functional diversity. Therefore, we next studied the effect of ectopic expression of α_2_δ subunits and Cachd1 on presynaptic calcium channel abundance in wildtype hippocampal neurons. Overexpression of α_2_δ-1 induced a more than two-fold increase in the intensity of presynaptic endogenous Ca_V_2.1 labelling compared to eGFP-only expressing control neurons (Figure 5A,B, eGFP and α_2_δ-1). This was in line with α_2_δ-1-mediated increases in Ca_V_2.1 currents upon co-expression [29], and is further supported by the previous observation that the expression of α_2_δ-1 in hippocampal neurons increased the synaptic labelling intensity of Ca_V_2.1 channels [30]. Contrary to α_2_δ-1, the overexpression of α_2_δ-4 or Cachd1 did not increase presynaptic clustering of endogenous Ca_V_2.1 (Figure 5A,B). The α_2_δ-1-induced increase of presynaptic Ca_V_2.1 was not caused by an overall effect of α_2_δ-1 on presynaptic bouton size, as no concomitant increase of synapsin labelling could be observed (Figure 5C). These results are consistent with previous observations showing the predominant interaction of α_2_δ-4 with Ca_V_1.4 [17] instead of Ca_V_2.1, and that Cachd1 had no effects on currents of heterologously expressed Ca_V_2.1 [24]. Because Cachd1 was previously shown to increase Ca_V_2.2 currents and cell surface expression upon heterologous co-expression [24], we next analyzed the synaptic expression levels of endogenous Ca_V_2.2. In comparison to control (eGFP-only) neurons, we observed neither significant increases in presynaptic Ca_V_2.2 nor synapsin abundance upon overexpression of α_2_δ-1, α_2_δ-4, and Cachd1 (Figure 6). Together, this shows that α_2_δ proteins differentially modulate the abundance of presynaptic calcium channels.

### 2.5. α_2_δ-4 and Cachd1 Do Not Affect Glutamatergic Synapse Composition

Because α_2_δ proteins differentially modulate the abundance of presynaptic calcium channels, and because specific α_2_δ isoforms and splice variants have previously been shown to strongly affect postsynaptic receptor abundance [10,31], we next sought to determine whether α_2_δ-1, α_2_δ-4, and Cachd1 would differentially modulate the molecular composition of glutamatergic synapses. Since the overexpression of a splice variant of α_2_δ-2 in excitatory glutamatergic neurons induces aberrant axonal wiring associated with a striking induction of mismatched synapses by recruiting exclusively postsynaptic GABA_A_-receptors subunits opposite glutamatergic presynaptic nerve terminals [11], we investigated the abundance of both glutamatergic and GABAergic proteins in glutamatergic synapses. To this end, we quantified the expression levels of pre- and postsynaptic proteins of excitatory glutamatergic (vGlut1, GluR1) and inhibitory GABAergic (vGAT, GABA_A_-R, Gephyrin) synapses upon overexpression of α_2_δ-1, α_2_δ-2-ΔE23, α_2_δ-4, and Cachd1, compared to eGFP-only as control. As previously demonstrated [11], the overexpression of an alternative splice isoform of α_2_δ-2 that lacks exon 23 (α_2_δ-2-ΔE23), and thereby forms an α-helix, induces the formation of mismatched synapses, identified by postsynaptic GABA_A_ receptor clusters opposite vGlut1 positive terminals (Figure 7A,C, α_2_δ-2). However, neither overexpression of α_2_δ-1 and α_2_δ-4 nor of Cachd1 resulted in changes in the molecular composition of glutamatergic synapses (Figure 7A–F). As was expected for glutamatergic synapses, the quantification of labelling intensities confirmed a strong expression of presynaptic vGlut1 (Figure 7B) and postsynaptic AMPA-receptors (GluR1, Figure 7E) in synapses overexpressing α_2_δ-1, α_2_δ-4, and Cachd1. In contrast, and except for α_2_δ-2 (see above), components of GABAergic synapses were not expressed (vGAT, GABA_A_-R, gephyrin in Figure 7C,D,F), as was expected due to the presence of either exon 23 or a corresponding structural feature (extra loop). Moreover, in contrast to α_2_δ-3 [11], presynaptic bouton sizes were not affected by overexpression of α_2_δ-1, α_2_δ-2, α_2_δ-4, and Cachd1 (Figure 7G).

### 2.6. Differential Effects on Presynaptic Calcium Transients

So far, our experiments have shown that α_2_δ-1, α_2_δ-4, and Cachd1 differentially affect Ca_V_2.1 channel density but have no effect on the molecular composition of glutamatergic synapses. Nevertheless, rescue experiments in α_2_δ TKO/KD neurons revealed subtle differences in the peak amplitudes of presynaptic calcium transients (Figure 4), even between conditions (α_2_δ-4 vs. Cachd1) showing similar effects on presynaptic calcium channel density. Hence, α_2_δ-1, α_2_δ-4, and Cachd1 may differentially modulate the function of presynaptic calcium channels beyond an exclusive effect on synaptic channel density. To gain insights into isoform differences in synapse-specific functions, we next analyzed the presynaptic calcium signals in wildtype neurons overexpressing the individual proteins. In this experimental setting, the overexpressed proteins compete with endogenously expressed α_2_δ subunits (α_2_δ-1 to -3) and Cachd1. We observed that α_2_δ-1, in contrast to a previous report [30], and Cachd1 overexpression led to a significant increase in the peak calcium amplitudes compared to control (1 AP) and α_2_δ-4 (1, 3, and 10 APs; Figure 8A). Overexpression of α_2_δ-4, however, resulted in the opposite effect, reducing calcium transients in response to all stimulation patterns (Figure 8A) in comparison to α_2_δ-1 and Cachd1 expressing neurons. These differences were noticeable in the mean traces (Figure 8A; ΔF/F0) and in the frequency distribution histogram (Figure 8B; left-shift of blue α_2_δ-4 curve along the X-axis), as well as in the means of all individual cells analyzed (Figure 8C). Taken together, our analysis of presynaptic calcium transients revealed differential effects of neurons over-expressing α_2_δ-1 and Cachd1 compared to α_2_δ-4. The observed effects are in line with the rescue potential of the severe synaptic phenotype in α_2_δ TKO/KD neurons (Figure 4). However, the effect on calcium transients cannot be fully explained by the modulation of the abundance of presynaptic Ca_V_2.1 and Ca_V_2.2 channels (Figure 5 and Figure 6), even though smaller transients in α_2_δ-4 expressing neurons match the slight, although not significant, tendency to reduce presynaptic calcium channel expression. 

### 2.7. Structural Determinants for Protein–Protein Interactions of α_2_δ Subunits and Cachd1 with Pore-Forming α_1_ Subunits

Our experiments have shown, on the one hand, that α_2_δ-1, α_2_δ-4, and Cachd1 can all rescue the severe synaptic defect in α_2_δ TKO/KD neurons, and on the other hand, that they differentially modulate presynaptic calcium channel abundance and calcium transients. Some of the effects on presynaptic calcium transients may be related to the effect on regulating channel density (e.g., α_2_δ-1). However, the positive (Cachd1) and negative (α_2_δ-4) effects on presynaptic calcium transients cannot be consistently explained by the regulation of the density of the most abundant presynaptic calcium channels Ca_V_2.1 and Ca_V_2.2. Hence, α_2_δ-4 and Cachd1 may preferentially associate with other pore-forming subunits or mediate a negative effect by competing with endogenous α_2_δ subunit partners.

To better understand the structural features that might account for differences in the functionality between distinct α_2_δ isoforms and Cachd1, we performed structural homology modelling between the pore-forming Ca_V_2.1 α_1_ subunit and α_2_δ-1, α_2_δ-4, and Cachd1. Using the previously published cryo-EM structure of human Ca_V_2.2 complexed with α_2_δ-1 as a reference (Figure 9A, [32]) we modelled the interaction between the predicted AlphaFold structures of Ca_V_2.1 with α_2_δ-1, α_2_δ-4, and Cachd1 (Figure 9B). Amino acid residues previously identified in the Ca_V_2.2 complex, which are important for α_1_–α_2_δ interactions, were further analyzed for their conservation and interaction potential in the obtained Ca_V_2.1 models. Ca_V_2.2 forms a total of eight hydrogen bonds with α_2_δ-1. The interacting residues in Ca_V_2.2 (Asp120, Asp122, Asn636, Phe637, Glu1314, and Arg1339) are conserved in Ca_V_2.1 (Asp125, Asp127, Asn643, Phe644, Glu1360, and Arg1385) and, importantly, are also predicted interaction sites in all modelled complexes with α_2_δ-1, α_2_δ-4, and Cachd1. When Ca_V_2.1 is complexed with α_2_δ-1, two additional hydrogen bonds between the backbone amides of Met131 and Pro124 of Ca_V_2.1 are expected to form with Glu366 and S263 of α_2_δ-1. Also, the models predicted one additional hydrogen bond between the backbone oxide of V1380 of Ca_V_2.1 and Lys221 of Cachd1. Taken together, there seems to be a conserved interaction pattern of Cav α_1_ subunits together with α_2_δ proteins. However, structural homology modelling suggests that the interaction strength of different α_1_-α_2_δ complexes may be fine-tuned by additional interaction sites (hydrogen bonds). Hence, the models for the specific interactions tested (exclusively α_2_δ-1, α_2_δ-4, and Cachd1 were investigated) predicted that Ca_V_2.1 would form the strongest interaction with α_2_δ-1, followed by Cachd1 and, finally, α_2_δ-4.

## 3. Discussion

Previously, it has been shown that the neuronal α_2_δ subunits α_2_δ-1, -2, and -3 regulate and modulate synapse formation, differentiation, and synaptic transmission [7,9,11,25]. Furthermore, the largely retinal α_2_δ-4 has been identified as an organizer of synaptic ribbons in photoreceptor cells [18]. Here, we have shown that regulating synapse formation and modulating synaptic functions is a general feature of all α_2_δ isoforms and of the α_2_δ-like protein Cachd1. First, α_2_δ-4 and Cachd1 can localize to presynaptic terminals in hippocampal neurons. Second, and similar to the most abundant neuronal α_2_δ proteins, the expression of both proteins can rescue the severe synaptic defect of α_2_δ TKO/KD neurons in terms of synapse differentiation, presynaptic calcium channel localization, and synaptic function. Third, rescuing presynaptic function by the expression of individual α_2_δ proteins resulted in subtle differences in the amplitude of presynaptic calcium transients. Similar differences were observed upon over-expression of these proteins in wildtype hippocampal neurons. These differences did not correlate with the α_2_δ-mediated increase in presynaptic calcium channel density. Hence, our data suggest that α_2_δ isoforms and Cachd1 differ in their ability to interact with presynaptic calcium channels. This was also reflected by distinctly matching charge distributions at the protein interaction surface between α_1_ and α_2_δ subunits and the number of predicted interaction sites. Taken together, all α_2_δ and α_2_δ-like proteins can regulate synapse formation, synapse differentiation, and synaptic function via a presynaptic mechanism. However, these effects can only partially be explained by calcium-channel dependent mechanisms.

### 3.1. Functional Redundancy of α_2_δ Proteins

To study the functions of individual channel subunits, it is pivotal to have a system that allows analyzing the effects of individual α_2_δ isoforms. However, this is complicated by the fact that hippocampal neurons simultaneously express three different isoforms [15]. Hence, here, we employed previously established α_2_δ subunit TKO/KD cultured hippocampal neurons as an expression system for α_2_δ isoforms and the α_2_δ-like protein Cachd1. α_2_δ TKO/KD neurons displayed a severe defect in presynaptic function and the formation and differentiation of glutamatergic synapses, as evidenced by strongly reduced presynaptic calcium transients, presynaptic calcium channel localization, presynaptic vesicle-associated proteins, and postsynaptic differentiation [25]. Expression of α_2_δ-4 and Cachd1 in TKO/KD neurons rescued the severe phenotype by restoring presynaptic Ca_V_2.1 channel localization (Figure 3) and calcium signaling (Figure 4). Hence, α_2_δ-4 and Cachd1 can mediate presynaptic synapse formation and differentiation to a similar degree as the most abundant neuronal isoform, α_2_δ-1. The observations that each α_2_δ subunit (α_2_δ-1, -2, -3; [25]) and α_2_δ-4 and Cachd1 (this study) can rescue glutamatergic synapse formation and function in TKO/KD neurons, and that they can locate to presynaptic nerve terminals (Figure 2), strongly suggests that α_2_δ proteins mediate redundant presynaptic functions. Principally, the rescue of presynaptic function could be explained either by calcium channel-dependent or calcium channel-independent mechanisms. The facts that presynaptic calcium transients could effectively be restored in α_2_δ TKO/KD neurons by all α_2_δ proteins, and that the residues involved in the interaction with the α_1_ subunit of Ca_V_2.1 were highly conserved between α_2_δ isoforms and Cachd1 (Figure 9) point to a channel-dependent mechanism. Furthermore, all expressed proteins also restored presynaptic Ca_V_2.1 localization in α_2_δ TKO/KD neurons. In contrast to the rescue experiments, the consequences of α_2_δ protein over-expression on presynaptic calcium transients and calcium channel localization in wildtype hippocampal neurons seem to be more complex. The over-expression of α_2_δ-1 strongly increased presynaptic Ca_V_2.1 abundance, while that of Cachd1 and α_2_δ-4 had no effect. Importantly, these differential effects on presynaptic Ca_V_2.1 clustering were not related to the observed differences in neuronal surface expression (confer Figure 1). Presynaptic Ca_V_2.2 expression, on the other hand, was not affected by any of the over-expressed proteins. Even though the over-expression of the proteins differentially modulated presynaptic calcium transients (see below), they altered neither the pre- and postsynaptic protein composition nor presynaptic bouton size (Figure 7). Together, this suggests that in addition to channel-dependent mechanisms, other channel-independent mechanisms are also involved in presynaptic functions. 

The involvement of channel-independent mechanisms is also supported by other recent observations. First, Cachd1 preferentially complexes with T-Type channels [22], while α_2_δ-1/-2/-3 are likely associated with N-, P/Q-, and L-type channels in the CNS. α_2_δ-4 is the primary partner of the L-type channel Ca_V_1.4 in retinal photoreceptor cells; however, its preferred α_1_ subunit in hippocampal neurons is elusive. Nevertheless, all investigated proteins were shown to be able to rescue presynaptic calcium transients (Figure 4) and Ca_V_2.1 channel localization (Figure 3). Second, our previous findings suggest that the trans-synaptic regulation of synaptic wiring and postsynaptic GABA_A_-receptor accumulation by α_2_δ-2 depends on the lack of alternatively spliced exon 23, but not on the number of presynaptic calcium channels [11]. This was supported by our present observation that none of the expressed proteins (which, in contrast to α_2_δ-2, all include exon 23 or a similarly organized protein sequence (Cachd1)) affected postsynaptic GABA_A_-receptor abundance. Third, we have previously shown that glutamatergic synapse formation in α_2_δ TKO/KD neurons could be rescued by mutant α_2_δ constructs lacking the MIDAS motif, which is necessary for calcium current enhancement and channel trafficking [25]. Finally, a recent study demonstrated that synapse assembly is independent of presynaptic Ca_V_2 channels and Ca^2+^ entry [33].

### 3.2. Differential Presynaptic Effects of α_2_δ-4 and Cachd1 

In contrast to their apparent redundancy in rescuing glutamatergic synapse formation and function (see above), we observed α_2_δ isoform-specific differences. First, α_2_δ isoforms displayed differences in their overall expression on the neuronal surface (Figure 1). Second, they differentially affected presynaptic calcium transients, both in the rescue (Figure 4) and the over-expression (Figure 8) condition, even though the observed differences were rather subtle. Third, α_2_δ subunits differed in their ability to increase presynaptic Ca_V_2.1 expression (Figure 5). Fourth, they differed slightly in their surface charges and the number of interacting residues with the α_1_ subunit (Figure 9).

All studied HA-tagged proteins are located in axons and synapses. Notably, Cachd1 and α_2_δ-4 also showed strong somato-dendritic surface expression (Figure 1C), which may reflect their preferential interaction with T-type or L-type channels, respectively. In presynaptic boutons, Cachd1 showed a similar distribution pattern to α_2_δ-1, whereas α_2_δ-4 was predominantly located in the perisynaptic membrane, as also observed for α_2_δ-2. Ultimately, and provided suitable antibodies or experimental tools are available, these expression patterns need to be confirmed by high resolution microscopy of the endogenously expressed proteins. α_2_δ-4 reduces presynaptic calcium transients upon over-expression in wildtype neurons (Figure 8) and shows a slightly reduced efficiency in rescuing the calcium transients in α_2_δ TKO/KD (Figure 4). In contrast, Cachd1 increases presynaptic calcium transients and rescues the TKO/KD phenotype, similarly to α_2_δ-1 (Figure 4 and Figure 8). Together, this may indicate that, like α_2_δ-1, Cachd1 specifically modulates active zone calcium channels upon over-expression, resulting in the observed increase of presynaptic calcium transients. In this context, it is noteworthy that Cachd1 can compete with α_2_δ-1 for binding to Ca_V_2.2 upon heterologous expression [24]. On the other hand, the rescue experiments in the TKO/KD neurons excluded competition with endogenous α_2_δ isoforms. In these experiments, Cachd1 rescued presynaptic calcium transients, like α_2_δ-1. The perisynaptic expression of α_2_δ-4 indicated a less likely interaction with active zone channels, resulting in smaller calcium signals in the rescue experiments. The dominant-negative effect of α_2_δ-4 on the calcium signal in the over-expression condition may either result from structural interference with the function of wildtype proteins in and around the active zone or from competition with endogenous α_2_δ subunits for presynaptic channels. These mechanisms may induce a shift in the contribution of the different presynaptic calcium channels to the calcium transients. This hypothesis can be tested in the future via the sequential application of specific calcium channel antagonists.

### 3.3. Synaptic Roles of α_2_δ-4 and Cachd1 

Our results provide a clear indication that, on the one hand, α_2_δ subunits are redundant regulators of synapse differentiation and function (see also [25]), and that, on the other hand, they differ in their modulation of presynaptic calcium channel abundance and calcium signaling. Until recently, α_2_δ-4 was considered to be an exclusive subunit of Ca_V_1.4 channels in retinal photoreceptor cells. However, mutations in CACNA2D4 in patients with neurodevelopmental disorders [34,35,36] and increased α_2_δ-4 mRNA expression levels in the hippocampus after status epilepticus [20] suggest previously unrecognized functions in the brain. This is further supported by the fact that α_2_δ-4 knockout mice exhibit behavioral impairments in terms of cognitive and motor functions [21]. Given the extremely low expression of α_2_δ-4 in the hippocampus [15], α_2_δ-4 might be restricted to a specific and very small subpopulation of neurons in the brain [5]. The present study supports the possibility of more diverse neuronal functions of α_2_δ-4, as it proves, for the first time, that it can indeed modulate synaptic and calcium-channel dependent functions in classical CNS neurons. Unlike the well-studied synaptic functions of α_2_δ-4 in the retina, the synaptic functions of Cachd1 remain unclear. However, Cachd1 has recently been described as an in vivo target for the Alzheimer’s protease BACE1 and potentially also a γ-secretase [37]. Considering our present findings, and that Cachd1 was identified as a modulator of Ca_V_3 activity [22], it may serve as a potential target in diseases such as epilepsy and pain. 

### 3.4. Conclusions

The present study, together with recent findings ([11,22,25] and others), suggests redundant critical and specific modulatory functions of α_2_δ proteins. Hence, the question arises: why does our nervous system express several distinct proteins with partially overlapping functions? Evolutionarily, it has been suggested that auxiliary subunits were gradually added to the calcium channel complex [38]. Most probably, they evolved to add to the complexity of electrical signaling. This seems to account for their specific and crucial regulatory function, as, contrary to other auxiliary subunits, α_2_δ proteins were not lost during evolution. It is hence tempting to speculate that the number of α_2_δ proteins increased to keep up with the complexity of neuronal signaling and synaptic transmission. Therefore, the fine-tuning of synaptic transmission may depend on the parallel action of more than one type of α_2_δ protein interacting with more than one calcium channel isoform in single synapses. Even if it is experimentally challenging from today’s perspective, revealing these parallel and specific functions will ultimately contribute to our understanding of the origin of neurological diseases, particularly neurodevelopmental disorders.

## 4. Materials and Methods

### 4.1. Breeding and Genotyping Procedures 

#### 4.1.1. Animals

All animal procedures for wildtype BALB/c and α_2_δ mutant mice were performed at the Medical University, Innsbruck, in compliance with government regulations and were approved by the Austrian Federal Ministry of Education, Science and Research (license number bmbwf 2020-0.107.333). Mice were maintained at the central animal facility in Innsbruck under standard housing conditions with food and water available ad libitum on a 12 h light/dark cycle.

#### 4.1.2. Breeding and Genotyping of Mutant Mice

WT and α_2_δ-2^+/du^ and α_2_δ-3^+/−^ mice were used for breeding procedures to obtain WT embryos and α_2_δ-2/3 double knockout p0-1 pups for primary neuronal cultures. The α_2_δ-3 knockout mice (α_2_δ-3^−/−^) (described in [11,25,31]) were generated by Deltagen (B6.129P2-Cacna2d3^tm1Dgen^; [39]), and α_2_δ-2^du/du^ mice, as described in [40,41], were purchased from The Jackson Laboratory (Bar Harbor, ME, USA) and bred and genotyped as previously described [11,25,31].

### 4.2. Cell Culture and Transfection Procedures 

#### Primary Cultures of Hippocampal Neurons for Fluorescence Imaging

Low-density hippocampal cultures were generated from 16.5- to 18-d-old embryonic BALB/c mice and p0-1 α_2_δ-2/3 double knockout mice as described previously [26,27,42]. Briefly, 2–8 hippocampi were dissected in cold Hank’s balanced salt solution (HBSS) following dissociation by 2.5% trypsin treatment and subsequent trituration. Dissociated neurons were plated at a density of ~100,000 cells/60 mm culture dish, on five 18 mm glass coverslips (#1.5; GML) coated with poly-l-lysine (Sigma-Aldrich, St. Louis, MO, USA). After attachment of neurons for 3–4 h, coverslips were transferred neuron-side down into a 60 mm culture dish containing a monolayer of glia. Neuros were maintained in serum-free neurobasal medium supplemented with Glutamax and B-27 (NBKO, all ingredients from Thermo Fisher Scientific, Waltham, MA, USA) changed weekly by a replacement of 1/3 of the volume. Glial proliferation was inhibited 3 days post plating with Ara-C (5 μM). On day in vitro (DIV) 6 neurons were transfected with plasmids using Lipofectamine 2000 (Thermo Fisher Scientific) as described previously [26]. The maximal DNA amount used for co-transfections totaled 1.5 μg of DNA at a 1:1:1 molar ratio. For presynaptic calcium measurements, neurons were used between DIV14 and 17, whereas immunolabelling experiments were conducted on neurons between DIV20 and 24.

### 4.3. Expression Vectors and Cloning Procedures

For live cell surface labelling, all α_2_δ subunit constructs and Cachd1 were tagged with a double HA-epitope at the N-terminus between the fourth and fifth amino acid after the predicted signal peptide cleavage site and for comparability cloned into a eukaryotic expression plasmid containing a neuronal chicken β-actin promoter, pβA, as described previously: pβA-α_2_δ-1 [11], pβA-2HA-α_2_δ-1 [11], pβA-α_2_δ-2-v1 [11], pβA-2HA-α_2_δ-2-v1 [11], pβA-α_2_δ-3 [11], pβA-eGFP-U6-shRNA-α_2_δ-1 [25], pHR-pβA-mCherry [11], pHR-pβA-mCherry-U6-shRNA-α_2_δ-1 [25], pSyn-GCaMP6f [28], pβA-eGFP [26,43].

**pβA-Cachd1:** The Cachd1 cDNA derived from mouse cortex was cloned into the pβA vector. Primer sequences were selected according to GenBank NM_198037.1. Briefly, the cDNA (nt 1–3867) of Cachd1 was amplified by PCR in three fragments: fragment 1 (nt 1–1366), fragment 2 (nt 1147–2892), and fragment 3 (nt 2707–3867). The forward primer used for amplifying fragment 1 introduced a HindIII site and the Kozak sequence (CCTACC) upstream the starting codon, while the reverse primer used for amplifying fragment 3 introduced a RsrII site after the stop codon. Fragments 1 and 2 were digested with HindIII/Bsu36I and Bsu36I/BamHI, respectively, and co-ligated into the corresponding HindIII/BamHI sites of the pβA vector, generating an intermediate construct. Fragment 3 was digested with BamHI/RsrII and finally ligated into the corresponding sites of the intermediate construct, yielding pβA-Cachd1.

**pβA-2HA-Cachd1:** Since it was predicted that the signal protein cleavage would be present at position 49 using SignalP 4.1, the 2HA tag followed by a TEV cleavage site was introduced into the pβA-Cachd1 between the third and fourth amino acid after the predicted signal peptide cleavage site. Briefly, the cDNA sequence of Cachd1 (nt 1–864) was PCR amplified with overlapping primers introducing the double HA tag and the TEV sequence in separate PCR reactions using pβA-Cachd1 as a template. The two separate PCR products were then used as templates for a final PCR reaction with flanking primers to connect the nucleotide sequences and the resulting fragment was then HindIII/SalI digested and ligated into the corresponding sites of pβA-Cachd1, resulting in a pβA-2HA-Cachd1 construct.

**pβA-α_2_δ-4**: The pIRES human CACNA2D4 cDNA encoding the α_2_δ-4 isoform was a gift from Niccolò Bacchi, [44]. In brief, the cDNA sequence encoding human α_2_δ-4 (EU832150) was first subcloned into the pβA vector [26,43] with two amino acid exchanges A to T at position 710 and E to G at position 1034. The cDNA sequence of α_2_δ-4 (nt 1–1565) was PCR amplified with a forward primer introducing a NotI site and the Kozak sequence (CCTACC) upstream of the starting codon. This fragment was then NotI/BglII digested and co-ligated with the BglII/SalI digested α_2_δ-4 cDNA and the NotI/SalI digested pβA vector, yielding pβA-α_2_δ4. 

**pβA-2HA-α_2_δ-4**: Since a putative signal peptide was not predicted for α_2_δ-4, we used Signal P 4.0 to discriminate signal peptides from transmembrane regions [45]. The strongest prediction was that the signal peptide would comprise residues 1–68; therefore, the 2HA tag was introduced into pβA-α_2_δ-4 after residue I71. Briefly, the cDNA sequence of α_2_δ-4 (nt 1–1565) was PCR amplified with overlapping primers introducing the double HA tag in separate PCR reactions using pβA-α_2_δ-4 as a template. The two separate PCR products were then used as templates for a final PCR reaction with flanking primers to connect the nucleotide sequences. The resulting fragment was then NotI/BglII digested and ligated into the corresponding sites of pβA-α_2_δ-4, yielding pβA-2HA-α_2_δ-4. 

Sequence integrity of all newly generated constructs was confirmed by sequencing (MWG Biotech, Martinsried, Germany).

### 4.4. High-Resolution Fluorescence Microscopy

Immunolabeling of permeabilized or live-cell-stained neurons was performed as described previously [11,25,26,27,43,46]. Neurons were fixed with 4% paraformaldehyde/4% sucrose (pF) in PBS for 20 min at room temperature, washed, and incubated for 30 min in 5% normal goat serum in PBS containing 0.2% bovine serum albumin (BSA) and 0.2% Triton X-100 (PBS/BSA/Triton), enabling membrane permeabilization. Primary antibodies (Table 1) were diluted in PBS/BSA/Triton, applied in a humidified chamber overnight at 4 °C and detected by fluorochrome-conjugated Alexa secondary antibodies incubated for 1 h at room temperature. For surface staining of HA-tagged α_2_δ constructs, transfected neurons were incubated with rat-anti-HA antibody in glia conditioned neurobasal medium for 10 min at 37 °C following rinsing in warm HBSS and fixation with pF for 10 min at room temperature. Subsequent washing and blocking steps as well as 1 h incubation with fluorochrome-conjugated secondary goat anti-rat Alexa Fluor 594 antibody were conducted with PBS and PBS/BSA, respectively. After washing and postfixation of cells in pF for 5 min, neurons were permeabilized by blocking with PBS/BSA/Triton and incubated with primary mouse-anti-synapsin antibody overnight at 4 °C and detected with goat-anti-mouse Alexa Fluor 350 antibody. Coverslips were mounted with DABCO (Agilent Technologies, Santa Clara, CA, USA) glycerol fluorescence mounting medium. Hippocampal cultures were typically viewed with an Olympus BX53 microscope (Olympus, Tokyo, Japan) using a 60 × 1.42 numerical aperture (NA) oil-immersion objective lens and 14-bit grayscale images were recorded with a cooled CCD camera (XM10; Olympus) using cellSens Dimension software (Olympus) and further analyzed in MetaMorph software (Molecular Devices, San Jose, CA, USA). Images of randomly selected well differentiated positively transfected cells and transfected axons forming synapses with untransfected dendrites were acquired with the same exposure and gain settings for all conditions within an experiment for quantification analysis. Overly saturated neurons (based on eGFP and mCherry levels in presynaptic calcium measurements) were excluded from analysis and only cells with medium to low eGFP or mCherry expression were selected for further analysis. Figures were assembled in Adobe Photoshop CS6 using linear adjustments to correct black level and contrast.

#### Antibodies

Details on antibodies used within this study are given in Table 1.

### 4.5. Image Analysis and Quantification 

#### 4.5.1. Colocalization of Synaptic Proteins 

To determine the synaptic localization of HA-tagged α_2_δ subunits and Cachd1, as well as of presynaptic (synapsin, vGAT, vGLUT1, Ca_V_2.1) and postsynaptic proteins (GABA_A_R subtypes, gephyrin, GLUR1), the “line scan” function was applied in MetaMorph (Molecular Devices) [47]. Average fluorescence intensities of respective signals (green-A488, blue-A350 and red-A594) were measured along a line of 3 μm length, followed by background subtraction, and plotted in Microsoft Excel.

#### 4.5.2. Quantification of Fluorescent Clusters in Single Boutons 

To analyze the effects of α_2_δ subunits or Cachd1 expression in glutamatergic synapses on synapse composition and protein expression, images from triple-fluorescence labeled neurons were acquired in the eGFP (green), GABA_A_R_α1_ (red), and vGlut1 (blue) channels as described [11]. Images were analyzed with a custom programmed and semi-automated MetaMorph journal (Molecular Devices), as described previously [11,25], with a slight modification. Briefly, eGFP and vGlut1 images were superimposed, eGFP/vGlut1-positive varicosities (putative glutamatergic synapses) were randomly chosen as regions of interest (ROIs), and a background region was selected for background subtraction. Axonal varicosities were defined as prominent swellings with higher fluorescence signals compared to the adjacent axonal stem after thresholding the image. Subsequently, GABA_A_R, and vGlut1 grayscale images were measured without thresholding to avoid potential cut-off of the fluorescent signal. By applying the “shrink region to fit” tool, automatic boundaries were drawn according to the threshold enabling only colocalized clusters to be analyzed and selected ROIs were then transferred from the eGFP image to the GABA_A_R and vGlut1 images to measure fluorescent intensities. For the individual synapses in each of the channels, the following parameters were detected in a blinded manner: eGFP threshold area as a measure for bouton size and average and integrated fluorescence intensities providing information on the size and intensity of clusters. In the same manner, the expression of synapsin and Ca_V_2.1 was conducted. 

#### 4.5.3. Quantification of Live Cell Surface Expression 

To measure the intensity of live cell-stained HA-tagged constructs, the fluorescent intensities were determined by measuring the fluorescent intensity along a thresholded axon, dendrite, and the cell soma. 

#### 4.5.4. Quantification Analysis

Analyses of all experiments were conducted with Microsoft Excel. Mean values of individual cells from two to three independent culture preparations were calculated and plotted in Graphpad Prism 8. The Ca_V_2.1 and synapsin values in the rescue experiment were additionally normalized to the rescue control and log-transformed.

### 4.6. Calcium Imaging

A GCaMP6f version coupled to synaptophysin driven by a synapsin promotor was applied as described in [25,28]. Calcium imaging was conducted as described previously [25]. Briefly, to examine presynaptic Ca^2+^ influx, primary neurons were transfected at DIV6 with pβA-SynGCaMP6f and, as indicated, additional plasmids (pHR-βA-mCherry-U6-shRNA-α_2_δ-1, pβA-mCherry, pβA-α_2_δ-1, pβA-α_2_δ-4, pβA-Cachd1) were co-transfected. Then, 8–11 days post transfection, coverslips were mounted in a recording chamber, placed on an inverted microscope (Olympus IX71, 60x, 1.42 NA PlanApo objective), and superfused at 1.0–1.5 mL/min with bath solution (temperature 32 °C), containing (in mM): NaCl 145, KCl 3, MgCl_2_ 1.5, CaCl_2_ 2, glucose 11, HEPES 10; pH 7.3 adjusted with NaOH. The bath solution contained 10 μM 6-cyano-7-nitroquinoxaline-2,3-dione (CNQX), 25 μM D, L-2-amino-5-phosphonovaleric acid (AP5), and 10 μM bicuculline to suppress postsynaptic signaling. All chemicals were obtained from Sigma (St. Louis, MO, USA), except for CNQX, AP5 and bicuculline (Tocris). As described [28], a stimulation electrode, built by two platinum wires of 10 mm length with 10 mm distance, was positioned with a micromanipulator (MPC-200, Sutter Instrument, Novato, CA, USA) and neurons were stimulated with 50 Hz trains of 1, 3 and 10 current pulses (1 ms, 55 mA). Ca^2+^ transients were visualized and recorded (20 ms exposure time, frame rate 50 Hz, 200 frames, binning 2: 0.215 μm per pixel) with a CMOS camera (Orca Flash4.0, Hamamatsu, Japan), using a halogen lamp light source (HXP 120) in the green channel (excitation: 470/40 nm, emission: 525/50 nm). Image recordings were controlled by Micromanager software (Visitron Systems, Puchheim, Germany). As a baseline reference, 50 frames were recorded before the stimulus train was triggered. 

#### Data Analysis

Recordings were quantified in FIJI/ImageJ (National Institute of Health, Bethesda, MA, USA) as described previously [25]. Briefly, 22 ROIs per recorded cell were drawn around mCherry positive presynaptic boutons using the plugin ‘Time Series Analyzer V3′ with an AutoROI diameter of 10 pixels. The regions were subsequently used in the green SynGCaMP6f recordings to measure the fluorescence changes, removing the background signal with the commonly used “Subtract Background” tool of ImageJ (employing a “rolling ball” algorithm with a radius of 20 pixels ≈ 4.3 μm). The mean of the four highest fluorescence pixels was calculated for each ROI at each frame by applying a self-made macro [28], and further analysis was done in Microsoft Excel. To obtain mean sample traces, the changes in fluorescence as ΔF/F0 were calculated for each ROI, and the 22 synapses per cell were averaged. Statistical analysis was performed on the maxima of cells in Graphpad Prism 8, as indicated. Cumulative frequency distributions were obtained by calculating the maximal response (ΔF/F0) by averaging five frames of the peak signal for every single synapse in Microsoft Excel. For each condition and stimulation, a cumulative histogram with defined class sizes was calculated, and values were normalized to the number of synapses per condition. Data were plotted as peak response (log) against the frequency (%) starting at 0.01, as signals below this threshold were indistinguishable from noise and, therefore, considered as non-responding.

### 4.7. Homology Modeling

To obtain homology models of the individual α_1_-α_2_δ complexes, the predicted AlphaFold models of mouse Ca_V_2.1 (AlphaFold identifier: AF-P97445-F1), mouse α_2_δ-1 (AF-O08532-F1), human α_2_δ-4 (AF-Q7Z3S7-F1), and mouse Cachd1 (AF-Q6PDJ1-F1) were superposed onto the published cryo-EM Ca_V_2.2 complex structure (PDB code: 7miy) in PyMOL (The PyMOL Molecular Graphics System, Version 2.3.2. Schrödinger, LLC, New York, NY, USA). Improper surface interactions within the complexes required minor remodeling of α_2_δ-4 and Cachd1, performed in Coot [48], whereby all bond restraints were retained. Complex interface areas and interactions were analyzed using the PDBePISA server [49] and PyMOL. The long intracellular loops of the Ca_V_2.1 α_1_ subunit were shortened for clearer illustrations. PyMOL was used to prepare the figures.

## Figures and Tables

**Figure 1 ijms-23-09885-f001:**
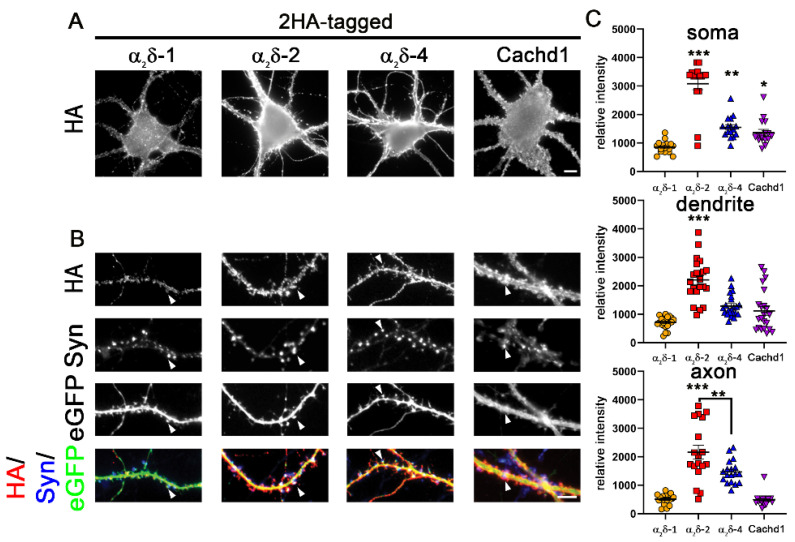
α_2_δ-4 and Cachd1 can be expressed on the somato-dendritic and axonal surface. (**A**) Live cell surface staining of wildtype primary hippocampal neurons overexpressing soluble eGFP together with HA-tagged constructs of α_2_δ subunits or Cachd1. (**A**) The somatic expression pattern of the anti-HA labelling revealed that all α_2_δ subunits and Cachd1 can be expressed on the neuronal surface. (**B**) Anti-HA and synapsin immunofluorescent labelling revealed that all α_2_δ subunits and Cachd1 show synaptic and dendritic membrane targeting (white arrowheads). Neurons overexpressing α_2_δ-1 displayed a weaker expression pattern on the somatic, dendritic, and axonal surface compared to α_2_δ-2, α_2_δ-4, and Cachd1. (**C**) Quantification of the average HA fluorescent intensities illustrated that the surface expression of α_2_δ-1 was reduced compared to those of the other subunits. α_2_δ-2 expression in soma, dendrites, and axons was higher compared to α_2_δ-1, α_2_δ-4, and Cachd1. Quantification data are shown as values for individual cells (dots) and means ± SEM. ANOVA with Tukey’s multiple comparison test (**C**) 16–22 cells per condition from three independent culture preparations. F_(3, 60)_ = 53.6, *p* < 0.0001. Significances of post hoc test (*** *p* < 0.001, ** *p* < 0.01, * *p* < 0.05) in comparison to α_2_δ-1, unless otherwise indicated, are indicated with asterisks. Scale bars, 5 µm.

**Figure 2 ijms-23-09885-f002:**
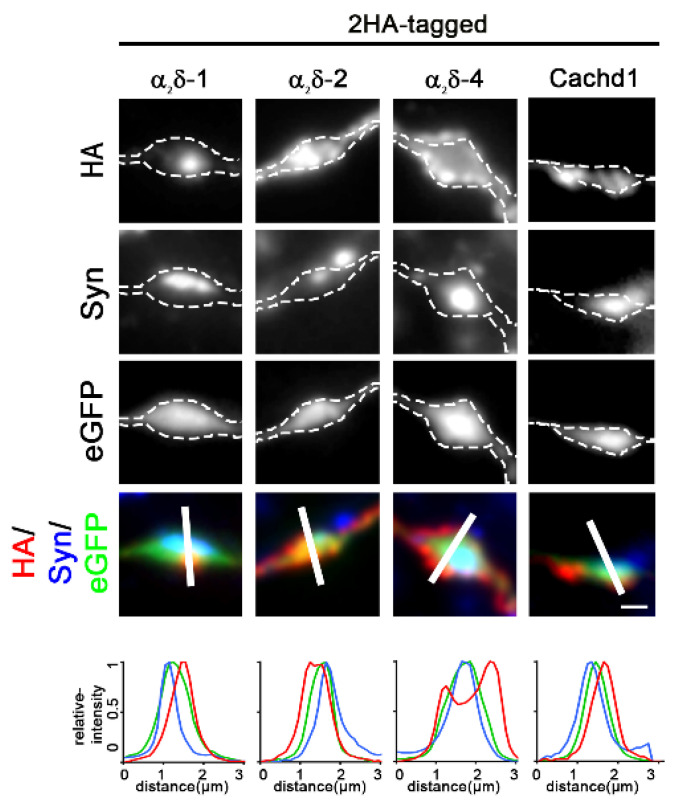
α_2_δ-4 and Cachd1 can target presynaptic membranes. An analysis of axonal varicosities illustrated a presynaptic localization pattern for all α_2_δ subunits and Cachd1. This can be inferred from the color overlay and the co-localization of the linescan peaks of HA-α_2_δ (red), synapsin (blue) and eGFP (green). Axonal varicosities were identified by their eGFP expression and are outlined by a dashed line. α_2_δ-4 specifically accumulated in the perisynaptic membrane around a central synapsin cluster. Overall, α_2_δ-2 and α_2_δ-4 displayed higher axonal expression compared to α_2_δ-1 and Cachd1 (for a quantitative analysis, see Figure 1C). Scale bar, 1 µm.

**Figure 3 ijms-23-09885-f003:**
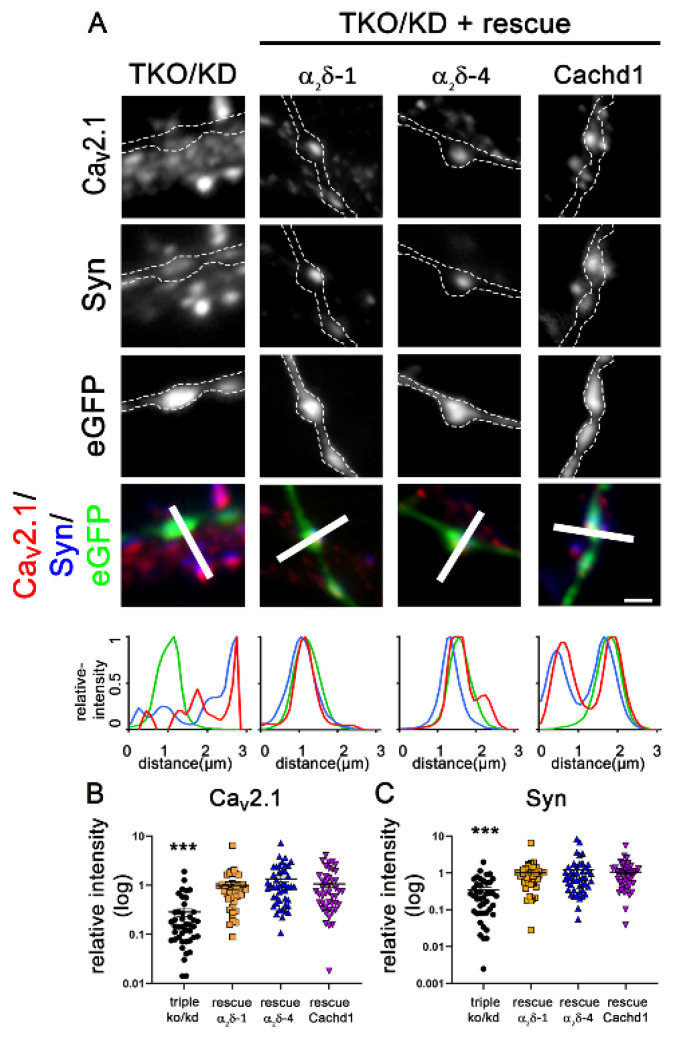
All α_2_δ subunits and Cachd1 rescue glutamatergic synapse formation. (**A**) Immunofluorescent labelling of synapsin and Ca_V_2.1 in α_2_δ TKO/KD neurons (α_2_δ-2/3 double knockout neurons transfected with shRNA-α_2_δ-1/eGFP) illustrated a severe defect in glutamatergic synapse formation and differentiation, the absence of synapsin and Ca_V_2.1 clusters in presynaptic varicosities. Boutons and axons were identified by their eGFP expression and are outlined by a dashed line. The reintroduction of α_2_δ-1, α_2_δ-4, or Cachd1 could rescue the accumulation of Ca_V_2.1 and synapsin in the presynaptic boutons. This is supported in the color overlay, qualitative linescan analysis, and the quantification of the fluorescent intensities of Ca_V_2.1 (**B**) and synapsin 1 (**C**). Quantification is represented as values for individual cells (dots) and means ± SEM. ANOVA with Tukey’s multiple comparison test with 39–46 cells per condition from three independent culture preparations. (**B**) F_(3, 169)_ = 25.85, *p* < 0.0001, post hoc: *** *p*< 0.0001. (**C**) F_(3, 169)_ = 19.25, *p* < 0.0001, post hoc: *** *p* < 0.0001. Significance of the TKO/KD in comparison to rescue conditions is indicated in the graphs with asterisks. Scale bar, 1 µm.

**Figure 4 ijms-23-09885-f004:**
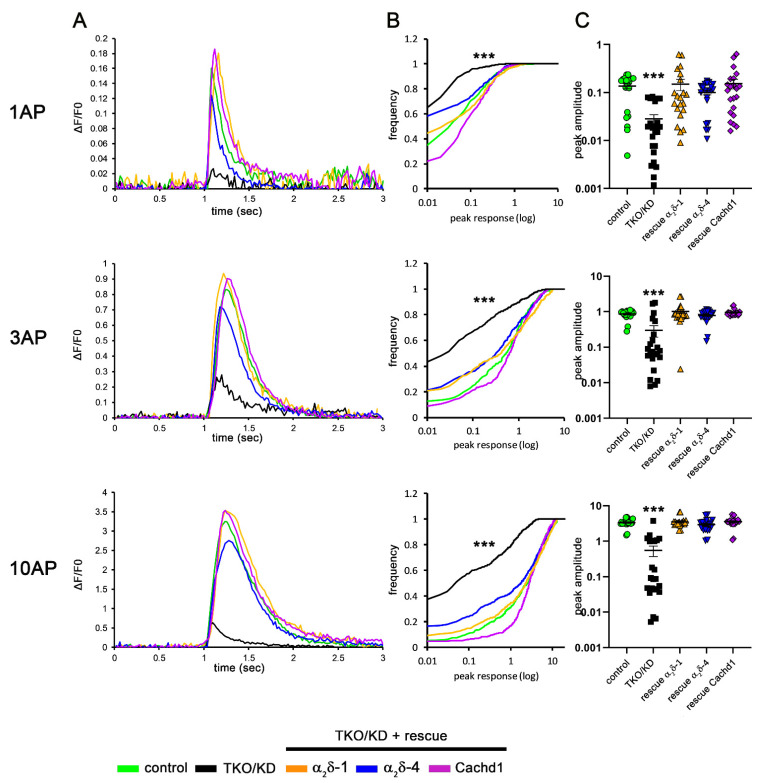
All α_2_δ subunits and Cachd1 can restore presynaptic calcium signaling in α_2_δ TKO/KD neurons. Mean sample traces (**A**), cumulative frequency distribution blots (**B**), and quantification (**C**) of presynaptic calcium imaging (SynGCaMP6f) in α_2_δ TKO/KD neurons. (**A**,**C**) Presynaptic calcium signals were significantly reduced in TKO/KD neurons (black) in response to 1 action potential (AP), 3 AP, and 10 AP stimulation compared to double heterozygous controls (green). The introduction of α_2_δ-1 (orange), α_2_δ-4 (blue), or Cachd1 (purple) restored presynaptic calcium transients in all stimulation patterns. (**B**) The fraction (see text) and mean of responding synapses in TKO/KD neurons was strongly reduced compared to control and rescue conditions. ANOVA with Tukey’s multiple comparison test: 1 AP: F_(4, 2504)_ = 30.7, *p* < 0.0001; 3 AP: F_(4, 2504)_ = 38.6, *p* < 0.0001; 10 AP: F_(4, 2504)_ = 95.0, *p* < 0.0001. Significances of post hoc tests compared to the TKO/KD condition are indicated in the graphs by asterisks (*** *p* < 0.001). (**C**) Quantification shows values for individual cells (dots) and means ± SEM. 21–23 cells per condition were obtained from three independent culture preparations (number of synapses analyzed: control, 483; TKO/KD, 506; rescue α_2_δ-1, 462; rescue α_2_δ-4, 484; rescue Cachd1, 484). ANOVA with Tukey’s multiple comparison test: 1 AP: F_(4, 105)_ = 4.5, *p* = 0.0021; 3 AP: F_(4, 105)_ = 11.6, *p* < 0.0001; 10 AP: F_(4, 105)_ = 35.9, *p* < 0.0001. Significances of post hoc tests of TKO/KD compared to the other conditions are indicated in the graphs by asterisks (*** *p* < 0.001).

**Figure 5 ijms-23-09885-f005:**
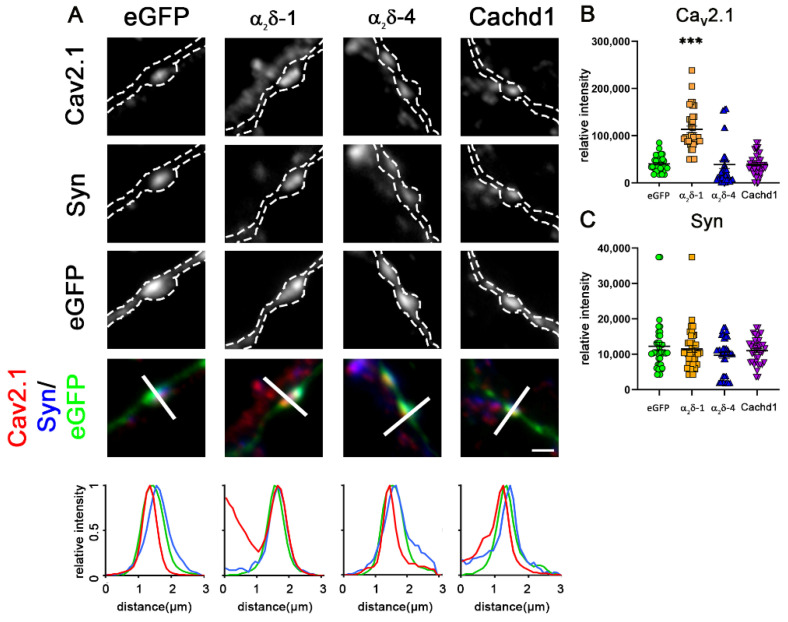
Overexpression of α_2_δ-4 or Cachd1 does not increase presynaptic clustering of Ca_V_2.1 channels. (**A**) Immunofluorescence analysis of axonal varicosities from wildtype neurons overexpressing eGFP and α_2_δ or Cachd1 constructs, labelled against synapsin and Ca_V_2.1. Micrographs show immunofluorescent signals of Ca_V_2.1 channels at presynaptic boutons, identified by eGFP expression (outlined with a dashed line) and presynaptic synapsin labelling along untransfected dendrites (see also qualitative linescan analysis). Contrary to neurons overexpressing α_2_δ-1, which display two times higher Ca_V_2.1 intensity, neurons overexpressing α_2_δ-4 and Cachd1 exhibited Ca_V_2.1 levels similar to those of the eGFP control. Quantification of Ca_V_2.1 (**B**) and synapsin 1 (**C**) shows values for individual cells (dots) and means ± SEM. Cells were obtained from three independent culture preparations. ANOVA with Tukey’s multiple comparison test. (**B**) 38 cells per condition, F_(3, 148)_ = 41, *p* < 0.0001. (**C**) 38 cells per condition, F_(3, 148)_ = 1.4, *p* = 0.25. Significance of post hoc test in comparison to α_2_δ-1 is indicated by asterisks (*** *p* < 0.001). Scale bar, 1 µm.

**Figure 6 ijms-23-09885-f006:**
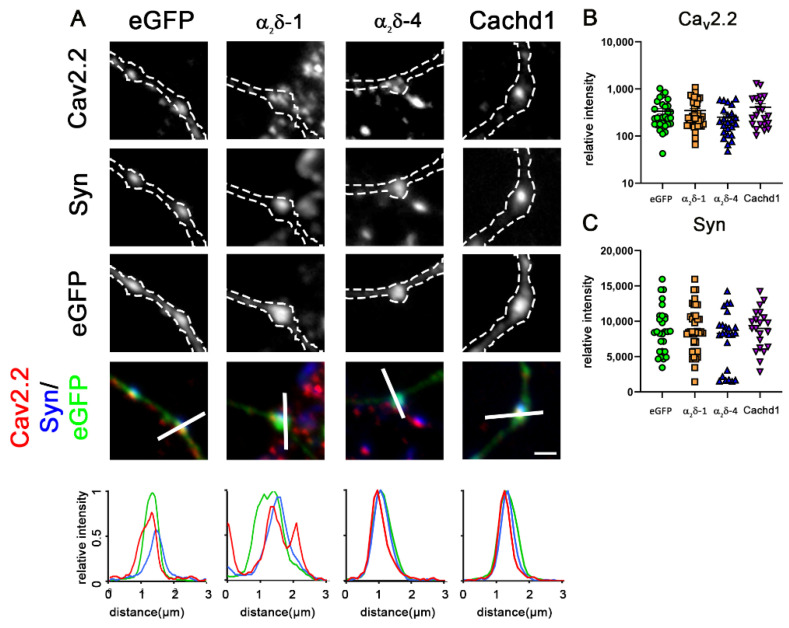
Overexpression of α_2_δ-1, α_2_δ-4, or Cachd1 does not increase clustering of N-type channels. (**A**) Immunofluorescence analysis of axonal varicosities from wildtype neurons overexpressing eGFP and α_2_δ or Cachd1 constructs, labelled against synapsin and Ca_V_2.2. Micrographs show immunofluorescent signals of Ca_V_2.2 channels at presynaptic boutons, identified by eGFP expression (outlined with a dashed line) and presynaptic synapsin labelling along untransfected dendrites (see also qualitative linescan analysis). Neurons overexpressing α_2_δ-1, α_2_δ-4, or Cachd1 exhibited Ca_V_2.2 levels similar to those of the eGFP control. Quantification of Ca_V_2.2 (**B**) and synapsin 1 (**C**) shows values for individual cells (dots) and means ± SEM. Cells were obtained from three independent culture preparations. ANOVA with Tukey’s multiple comparison test. (**B**) 24–39 cells per condition, F_(3, 110)_ = 1.58, *p* = 0.20. (**C**) 24–39 cells per condition, F_(3, 110)_ = 0.99, *p* = 0.40. Scale bar, 1 µm.

**Figure 7 ijms-23-09885-f007:**
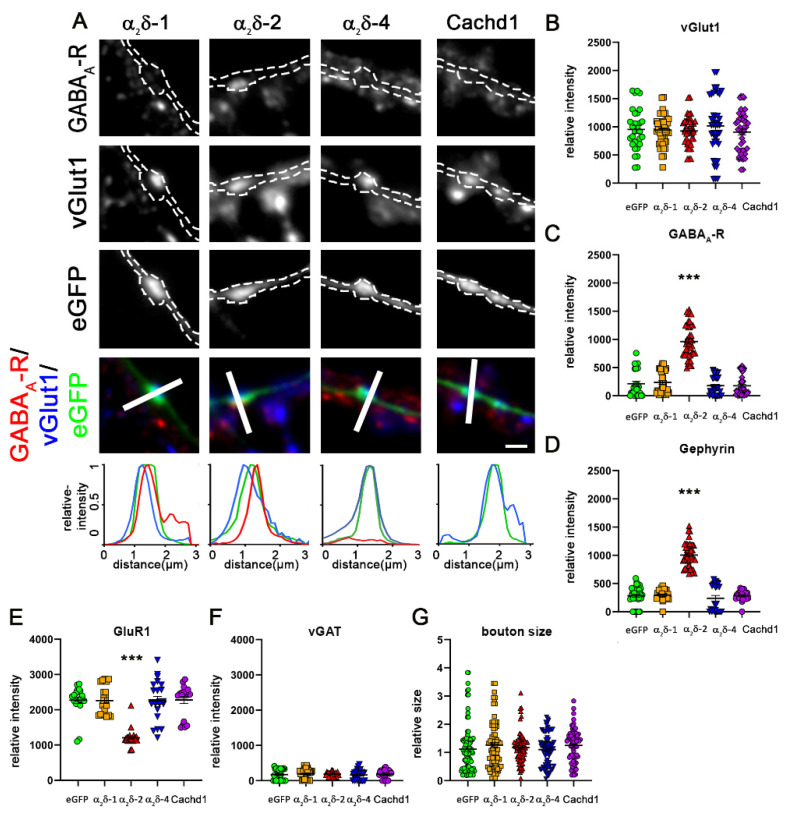
Overexpression of α_2_δ-4 or Cachd1 does not alter the composition of glutamatergic synapses. (**A**) Co-transfection of soluble eGFP and α_2_δ subunits or Cachd1 in combination with immunofluorescent labelling of presynaptic vGlut1 and postsynaptic GABA_A_-receptors was used to detect the formation of mismatched synapses. Only the overexpression of α_2_δ-2 led to the formation of mismatched synapses, as detected by postsynaptic GABA_A_ receptor clusters opposite vGlut1 positive glutamatergic terminals (A, α_2_δ-2). Overexpression of α_2_δ-4 or Cachd1 did not induce the formation of mismatched synapses and did not alter the molecular composition of glutamatergic synapses. Quantifications of immunofluorescence intensities of vGlut1 (**B**), GABA_A_ receptor (**C**), gephyrin (**D**), GluR1 (**E**), vGAT (**F**) and the bouton size as identified by eGFP fluorescence area (**G**) show values for individual cells (dots) and means ± SEM. Cells were obtained from three independent culture preparations. ANOVA with Tukey’s multiple comparison test was performed on 29–45 cells per condition. (**B**) F_(4, 203)_ = 0.58, *p* = 0.68. (**C**) F_(4, 203)_ = 103.6, *p* < 0.0001. (**D**) F_(3, 81)_ = 135.2, *p* < 0.0001. (**E**) F_(3, 108)_ = 28.8, *p* < 0.0001. (**F**) F_(3, 104)_ = 0.30, *p* = 0.88. (**G**) F_(3, 306)_ = 1.0, *p* = 0.51. Significances of post hoc test in comparison to α_2_δ-2 are indicated by asterisks (*** *p* < 0.0001). Scale bar, 1 µm.

**Figure 8 ijms-23-09885-f008:**
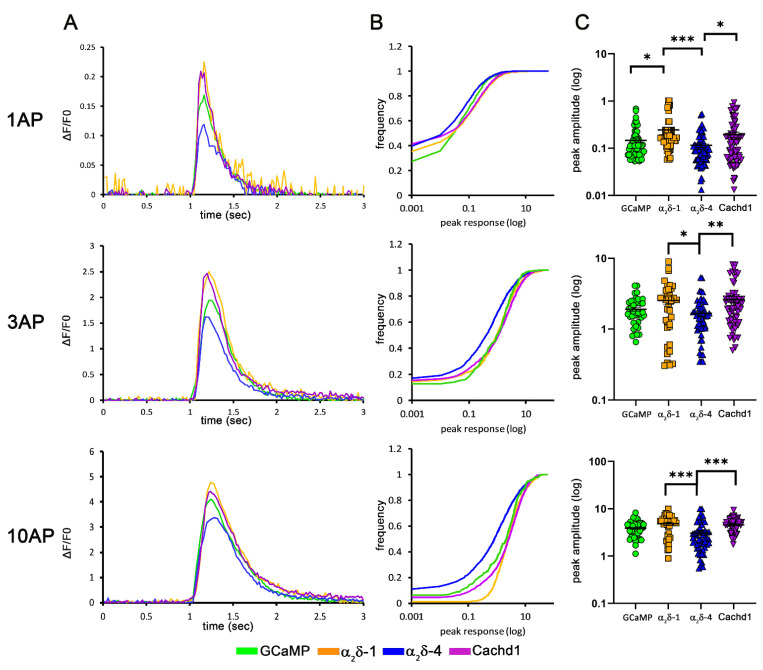
α_2_δ subunits and Cachd1 differentially affect presynaptic calcium transients. Mean sample traces (**A**), cumulative frequency distribution blots (**B**), and quantification (**C**) of presynaptic calcium signals (SynGCaMP6f) in wildtype neurons overexpressing either α_2_δ or Cachd1 constructs. Overexpression of α_2_δ-4 led to a reduction in presynaptic calcium signals in response to stimulation with 1 AP, 3 AP and 10 AP (blue) compared to control (green). In contrast, overexpression of α_2_δ-1 (orange) and Cachd1 (purple) increased calcium transients in all stimulation paradigms, as indicated in the mean sample traces and the maximal responses. Quantification shows values for individual cells (dots) and means ± SEM. 37–53 cells were obtained from four independent culture preparations (number of synapses analyzed: SynGCaMP6f, 1257; α_2_δ-1, 1122; α_2_δ-4, 1806; Cachd1, 1650). ANOVA with Tukey’s multiple comparison test: 1 AP: F_(3, 244)_ = 6.4, *p* = 0.0004; 3 AP: F_(3, 185)_ = 4.7, *p* = 0.0036; 10 AP: F_(3, 183)_ = 8.7, *p* < 0.0001. Significances of post hoc tests between conditions are indicated in the graphs by asterisks (*** *p* < 0.001, ** *p* < 0.01, * *p* < 0.05).

**Figure 9 ijms-23-09885-f009:**
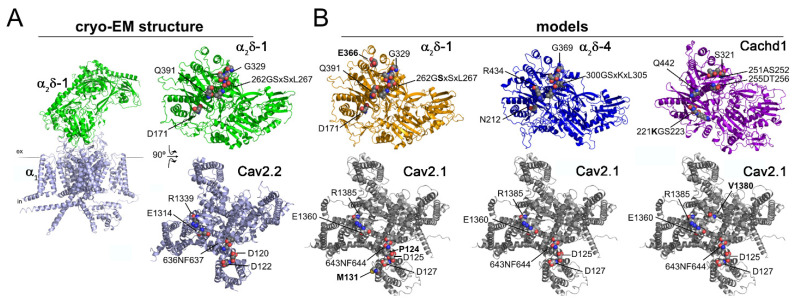
Structural interface analysis revealed preferences in complex formation. (**A**) Cryo-EM structure of the human Ca_V_2.2 calcium channel complex (PDB code: 7MIY), with the α_1_ subunit (light blue) and α_2_δ-1 (green.) Next to the membrane view, the complex is opened up to show the interaction interface between α_2_δ-1 (top) and α_1_ (bottom). Residues engaging in the subunit interactions are labelled and shown in sphere representation with the respective charges (blue, positive and red, negative). (**B**) Models of opened complexes (α_2_δ-1, top and α_1_, bottom) of the Ca_V_2.1 α_1_ subunit with mouse α_2_δ-1, human α_2_δ-4, and mouse Cachd1. The predicted interacting residues are labelled and shown in sphere representation in the respective charges (blue, positive, and red, negative); additional interacting residues are labelled in bold. Refer to text for description.

**Table 1 ijms-23-09885-t001:** List of antibodies.

Antibody	Dilution	Source
Anti-HA	1:100 (LIVE/A594)	Roche (Mannheim, Germany) (catalog #11867423001)
Anti-GABA_A_R_β2/3_	1:500 (A594)	Millipore (Darmstadt, Germany) (catalog #MAB341)
	1:250 (A350)	Millipore (catalog #MAB341)
Anti-gephyrin	1:2000 (A594)	Synaptic Systems (Göttingen, Germany) (catalog #147 021)
Anti-GLUR1	1:1000 (A594)	Upstate (Lake Placid, NY, USA) (catalog #06-306)
Anti-PSD-95	1:1000 (A594)	Thermo Fisher Scientific (Waltham, MA, USA) (catalog #MA1-045)
Anti-synapsin1	1:500 (A350)	Synaptic Systems (catalog #106 011)
Anti-vGLUT1	1:2000 (A350)	Synaptic Systems (catalog #135 002)
Anti-vGLUT1	1:500 (A594)	Synaptic Systems (catalog #135 511)
	1:250 (A350)	Synaptic Systems (catalog #135 511)
Anti-vGAT	1:500 (A350)	Synaptic Systems (catalog #131 002)
Anti-Cav2.1	1:2000 (A594)	Synaptic Systems (catalog #152203)
Anti-Cav2.2	1:2000 (A594)	Synaptic Systems (catalog #152313)
Goat anti-Mouse IgG, Alexa Fluor 350	1:500	Thermo Fisher Scientific (catalog #A-21049)
Goat anti-Rabbit IgG, Alexa Fluor 350	1:500	Thermo Fisher Scient (catalog #A-21068)
Goat anti-Mouse IgG, Alexa Fluor 594	1:4000	Thermo Fisher Scientific (catalog #A-11032)
Goat anti-Rabbit IgG, Alexa Fluor 594	1:4000	Thermo Fisher Scientific (catalog #A-11037)
Goat anti-Rat IgG, Alexa Fluor 594	1:4000	Thermo Fisher Scientific (catalog #A-11007)

## Data Availability

Data is contained within the article and raw data presented in this study are available upon request.

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
