# Peer review of "α_2_δ-4 and Cachd1 Proteins Are Regulators of Presynaptic Functions"

_ijms, 2022, doi:10.3390/ijms23179885_

Round 1

Reviewer 1 Report

This work is an interesting sequel to a series of works in which the authors have shown experimental evidence that the auxiliary subunits of the calcium channel complex Cava2d-1 to Cava2d3 contribute to determining the structure and function of the synapse. This work shows that two other proteins of this family (Cava2d-4 and Cachd1) can also participate in these events in hypothalamic neurons, basically using advanced microscopy and calcium imaging techniques in a triple KO murine model. The work is well planned and adequately presented, and the results support the conclusions reached by the authors.
My comments refer to the following general points:
1. It is known that the a2d-1 subunit increases the membrane expression of the AMPA receptor. However, in Figure 7E, it seems that the subunit has no effect relative to the control or a2d-4 and Cachd1. What could be the reason for this discrepancy?
2. On the other hand, a2d-2 decrease seems to have a significant effect on the functional expression of GABAR in the synapse. Could this be related to the presence of extrasynaptic GABARs? These receptors play an essential role in the excitability of neurons. Perhaps it would be worthwhile for the authors to discuss this point.
Minor point: 1. lines 118-119: improve the wording. A word is missing …” expression pattern of the revealed that”…

Author Response

Note: The line numbers below refer to the final manuscript after accepting all changes.

Reviewer 1:

  1. It is known that the a2d-1 subunit increases the membrane expression of the AMPA receptor. However, in Figure 7E, it seems that the subunit has no effect relative to the control or a2d-4 and Cachd1. What could be the reason for this discrepancy?

We think that the reviewer is referring to the work of Hui-Lin Pan’s group, which investigated the interaction between α2δ-1 and AMPA receptor using heterologous expression (Li et al., 2021) and synaptosomal lysates of spontaneously hypertensive rats (Zhou et al., 2022). These studies show that nerve injury potentiates postsynaptic, but not presynaptic, AMPARs in the spinal dorsal horn via α2δ-1 and that in spontaneously hypertensive rats this interaction between α2δ-1 and GluA1 is increased.

In our present study, however, we investigated presynaptic effects of α2δ subunits in cultured hippocampal neurons from wildtype (presumably normotensive) mice. Hence, in our experimental setting we analyzed trans-synaptic effects of presynaptically expressed α2δ proteins on postsynaptic receptors (e.g., GABAA-R or AMPA-R). Our study, therefore, does not exclude the possibility that somatodendritic/postsynaptic α2δ proteins may show additional protein-protein interactions (such as with the AMPA-R).

  1. On the other hand, a2d-2 decrease seems to have a significant effect on the functional expression of GABAR in the synapse. Could this be related to the presence of extrasynaptic GABARs? These receptors play an essential role in the excitability of neurons. Perhaps it would be worthwhile for the authors to discuss this point.

We assume the reviewer is referring to the quantification of Figure 7, in which we observed a strongly increased postsynaptic GABAA receptor expression (7C) upon overexpression of α2δ-2, which is concomitant with a decrease of GluR1 expression (7E). This finding is consistent with our previous study showing that presynaptic expression of α2δ-2-delta-e23 is crucial of the postsynaptic upregulation of GABAA receptors in glutamatergic synapses (Geisler et al., 2019; discussed in the present manuscript in the results, lines 299-310) and the discussion (lines 471-474). As suggested by the reviewer it is, in principle, possible that GABAA receptors are recruited by lateral diffusion from the extrasynaptic pool of receptors. Alternatively, synaptic GABAA receptor subunits may be added to the somatodendritic surface by exocytosis. In our previous study (Geisler et al., 2019, Fig. 4) we could show that only synaptic, but not extra-synaptic GABAA receptor subunits are recruited by presynaptic α2δ-2. This information has now been added to the results, lines 293-294.

Minor point: 1. lines 118-119: improve the wording. A word is missing …” expression pattern of the revealed that”…

Thank you for pointing out this oversight. We corrected the sentence to “The somatic expression pattern of the anti-HA labelling revealed that all α2δ subunits and Cachd1 can be expressed on the neuronal surface.”.

Reviewer 2 Report

By Using a combination of molecular biological, immunohistochemical labeling, Ca2+ imaging approaches as well as homology docking modelling, Albinger et al showed in cultured hippocampal neurons that α2δ-4 and Cachd1 can be targeted to presynaptic terminals and rescue subsynaptic defects in triple knockout/knockdown neurons lack of α2δ-1-3 isoforms, indicating that  α2δ isoforms are redundant in supporting synaptic connectivity and function. However, isoform-specific effects of over-expression in WT background, albeit subtle, diverge in regulating the abundance and types of presynaptic calcium channels, Ca2+ transients and postsynaptic excitatory/inhibitory receptors.These experimental results were further interpreted by analyzing the interface molecular interactions between α1and α2δ subunits, providing potential mechanistic explanations for their functional divergence in potentially regulating synaptic transmission.

Overall, this work is well done by employing multidisciplinary approaches. The text narrative is clearly presented in a logic manner. The findings are very solid, comprehensive and important for our general understanding of diverse functions of the α2δ family.

There are a couple of issues that the author may want to consider in order to boost the novelty and physiological relevance of this work.

1. There appears to be a lack of experimental evidence on whether or not there is any endogenous expression of  α2δ-4 and/or Cachd1 in WT hippocampal neurons in culture (or slices)? If there is, does introduction of HA-tagged construct lead to share similar expression patterns in different compartments of the neuron (Figure 1) as endogenous one?

2. The finding that in contrast to α2δ-1, α2δ-4/Cachd1 OE seems to decrease Ca2+ transients is very novel and interesting, but not explored sufficiently. It would be highly desirable that N- or P/Q-type toxins and blockers for other Ca2+ channels be tested here to decipher if Ca2+ transients are biased by altered expression of specific types of Ca2+ channels as a result of different α2δ/Cachd subunit.

3. The differences in the peak amplitude of presynaptic Ca2+ transients may be complemented by comparisons of the integrated area under each curve (Figure 8). It is interesting that 10 APs seems to make the differences in Ca2+ transients seen with 1 AP smaller in α2δ-4/Cachd OE synapses. Could these be interpreted an activity-dependent recruitment/activation of peri-synaptic calcium channels that are not active by 1AP?

Minor:

There are some typos that the authors need to do spelling check and correct.

Author Response

Note: The line numbers below refer to the final manuscript after accepting all changes.

Reviewer 2:

  1. There appears to be a lack of experimental evidence on whether or not there is any endogenous expression of α2δ-4 and/or Cachd1 in WT hippocampal neurons in culture (or slices)? If there is, does introduction of HA-tagged construct lead to share similar expression patterns in different compartments of the neuron (Figure 1) as endogenous one?

The reviewer correctly points out that currently only little information on the endogenous expression of α2δ-4 and Cachd1 is available. Cottrell et al., (2018) showed that endogenous Cachd1 is expressed strongly in the thalamus, cerebellum, and hippocampus. Moreover, as stated in the discussion (lines 526-528) we could demonstrate that α2δ-4 is expressed in the hippocampus at very low levels (Schlick et al., 2010). This has recently been supported in human temporal lobe epilepsy biopsies and mouse models of temporal lobe epilepsy, showing that mRNA levels of α2δ-4 increase in the hippocampus upon status epilepticus (Van loo et al., 2018). However, the precise subcellular expression pattern of both, Cachd1 and α2δ-4, is not yet known. Immunohistochemistry (Cottrell et al., 2018) demonstrated a strong cytoplasmic labelling of neurons, including dendrites of Cachd1, which is consistent with our HA labelling results. Experimental evidence regarding the subcellular localization of α2δ-4 in neurons is entirely missing. This is due on the one hand to a lack of α2δ-4 antibodies suitable for immunocytochemical analysis, and on the other hand the extremely low expression of α2δ-4 in hippocampus (see also discussion lines 526-528). The latter may indicate either very low levels in all hippocampal neurons (Klomp et al., 2022), or an expression in a specific and not yet identified subpopulation of hippocampal neurons (Schlick et al., 2010; Geisler et al., 2015; Ablinger et al., 2020). Taken together, currently extracellular HA-epitope labelling is the most reliable method for the comparative analysis of the subcellular distribution of these proteins.

Hence, to be more precise about the expression of Cachd1 and α2δ-4 in wildtype hippocampus we changed the wording in the results and discussion (lines 99-101 and 526-528) and now also discuss that the HA-labeling data ultimately need to be confirmed by high resolution analysis of the localization of the endogenous proteins (lines 496-498).

  1. The finding that in contrast to α2δ-1, α2δ-4/Cachd1 OE seems to decrease Ca2+ transients is very novel and interesting, but not explored sufficiently. It would be highly desirable that N- or P/Q-type toxins and blockers for other Ca2+ channels be tested here to decipher if Ca2+ transients are biased by altered expression of specific types of Ca2+ channels as a result of different α2δ/Cachd subunit.

α2δ-4 and Cachd1 actually behaved differently in our overexpression experiment (Fig. 8) because only α2δ-4 decreased calcium transients in all three stimulation conditions (1, 3, and 10 AP; blue lines and symbols in Fig. 8), whereas Cachd1 (magenta lines and symbols in Fig. 8) behaved more like α2δ-1. This highlights the importance of our overexpression experiments: Overexpression, presumably by competition with endogenous isoforms, reveals differences between α2δ-4 and Cachd1 that were not apparent in the rescue experiment (Fig. 4) in which both had a similar effect. This difference within the α2δ family is novel and interesting, as the reviewer recognizes. The reason for this differential behavior is yet unclear and has to remain open at present because the modulation of VGCC-mediated calcium influx by α2δ, in general, is not fully understood.

As to the suggestion of VGCC blocker experiments, we agree with the reviewer that the application of blocker toxins may shed light on the contribution of different VGCC subtypes to the calcium transients. However, a previous study by Tim Ryan’s group already demonstrated that overexpression of different α2δ isoforms does not lead to a significant shift in VGCC subtypes at nerve terminals (Hoppa et al., 2012). Moreover, obtaining conclusive novel data in such an experiment would require the sequential application of a series of blockers in conditions comparing all four α2δ subunits and Cachd1. Also, to exclude the possibility that the order of toxins affects the conclusion about the contribution of the different calcium channels, these experiments would have to be repeated with different application protocols. We plan to do such experiments in the future to confirm or refute the earlier observation for α2δ-4 and Cachd1, but this will require many months and a major effort that goes well beyond the scope of the present manuscript. To pay due respect to the consideration suggested by the reviewer, we included this point in the discussion (p16, lines 513-516).

  1. The differences in the peak amplitude of presynaptic Ca2+ transients may be complemented by comparisons of the integrated area under each curve (Figure 8). It is interesting that 10 APs seems to make the differences in Ca2+ transients seen with 1 AP smaller in α2δ-4/Cachd OE synapses. Could these be interpreted an activity-dependent recruitment/activation of peri-synaptic calcium channels that are not active by 1AP?

The reviewer raises interesting ideas here that would be applicable if the concentration of free calcium was measured, for example by Fluo5 or similar means. However, as in many other current studies, we used a genetically encoded calcium indicator of the GCaMP family (for the description of GCaMP6f see Chen et al., 2013), which has many advantages but is inherently limited by the (kinetic) properties of the indicator protein itself. Therefore, the integral under the transient curve mostly depends on the calmodulin moiety of the indicator that fails to release the bound calcium fast enough as evidenced by the fact that the fluorescence has not yet reached baseline levels even after 1.5 sec, long after the actual free calcium concentration in the terminal should have dropped again after stimulation. Thus, analysis of peak responses provides a more reliable and proportional readout of the calcium signal between experimental conditions based on the literature and our own experience (see for example Chen et al., 2013; Brockhaus et al., 2018; Brockhaus et al., 2019; Schöpf et al., 2021). Along this line, the smaller relative differences observed in 10 AP compared to 1 AP also depend on the property of the indicator because the limited number of GCaMP6f molecules expressed in a terminal become quickly saturated at repetitive stimulation.

Minor:

There are some typos that the authors need to do spelling check and correct.

As suggested, we did spell checks and corrected typos where necessary.

Round 2

Reviewer 2 Report

The authors have addressed my comments adequately with text revisions and made arguments for not pursuing additional experiments, which are acceptable.